# Context-Dependent Regulation of Peripheral Nerve Abundance by the PI3K Pathway in the Tumor Microenvironment of Head and Neck Squamous Cell Carcinoma

**DOI:** 10.3390/cells13121033

**Published:** 2024-06-14

**Authors:** Karam Khorani, Sebastian Burkart, Christopher Weusthof, Rui Han, Siyuan Liang, Fabian Stögbauer, Jochen Hess

**Affiliations:** 1Department of Otorhinolaryngology, Head and Neck Surgery, Heidelberg University Hospital, 69120 Heidelberg, Germany; sebastian.burkart@med.uni-heidelberg.de (S.B.); c.weusthof@stud.uni-heidelberg.de (C.W.); rui.han@stud.uni-heidelberg.de (R.H.); siyuan.liang@stud.uni-heidelberg.de (S.L.); 2Tissue Bank of the National Center for Tumor Diseases (NCT) Heidelberg, Germany and Institute of Pathology, Heidelberg University Hospital, 69120 Heidelberg, Germany; fabian.stoegbauer@tum.de; 3Institute of Pathology, School of Medicine, Technical University of Munich (TUM), 80337 Munich, Germany

**Keywords:** axonogenesis, head and neck squamous cell carcinoma, human papillomavirus, neurogenesis, perineural invasion, Schwann cells, tumor microenvironment, cancer–neuron interaction

## Abstract

Recent studies have highlighted neurons and their associated Schwann cells (SCs) as key regulators of cancer development. However, the mode of their interaction with tumor cells or other components of the tumor microenvironment (TME) remains elusive. We established an SC-related 43-gene set as a surrogate for peripheral nerves in the TME. Head and neck squamous cell carcinoma (HNSCC) from The Cancer Genome Atlas (TCGA) were classified into low, intermediate and high SC score groups based on the expression of this gene set. Perineural invasion (PNI) and TGF-β signaling were hallmarks of SC^high^ tumors, whereas SC^low^ tumors were enriched for HPV16-positive OPSCC and higher PI3K-MTOR activity. The latter activity was partially explained by a higher frequency of *PTEN* mutation and *PIK3CA* copy number gain. The inverse association between PI3K-MTOR activity and peripheral nerve abundance was context-dependent and influenced by the *TP53* mutation status. An in silico drug screening approach highlighted the potential vulnerabilities of HNSCC with variable SC scores and predicted a higher sensitivity of SC^low^ tumors to DNA topoisomerase inhibitors. In conclusion, we have established a tool for assessing peripheral nerve abundance in the TME and provided new clinical and biological insights into their regulation. This knowledge may pave the way for new therapeutic strategies and impart proof of concept in appropriate preclinical models.

## 1. Introduction

The tumor microenvironment (TME) is an ecosystem composed of stromal cells, immune cells, endothelial and lymphatic vasculature, and extracellular matrix components. Accumulating evidence indicates that these cellular and non-cellular components of the TME have a strong influence on tumor initiation, growth, invasion, metastasis, and response to therapy [1]. Recently, peripheral nerves have emerged as a novel pathological structure within the TME. Several studies have demonstrated the critical role of tumor innervation and signaling via neurotrophic factors and neurotransmitters in various types of cancer, including head and neck cancer [2,3,4]. For example, perineural invasion (PNI) has been identified as a common interaction between tumor cells and peripheral nerves, describing tumor invasion of existing locoregional nerves. PNI is commonly observed in cancers of nerve-rich regions such as prostate, pancreatic, and head and neck cancers. It is associated with cancer pain and serves as an independent risk factor for poor prognosis [5]. However, the cancer–neuron interaction (CNI) appears to be much more complex. Tumor cells release multiple factors, such as axon guidance molecules and nerve growth factors, to promote neurogenesis, axonogenesis, and cellular transdifferentiation into neuron-like cells [4,6,7]. These processes increase local nerve density, facilitate tumor progression, and are associated with poor prognosis [8,9].

Schwann cells (SCs) are the major supporting glial cells of the peripheral nervous system [10]. As the key component in neural repair and regeneration, SCs play a crucial role in promoting PNI [11,12]. Nerves invaded and damaged by tumor cells can activate SCs to proliferate and migrate toward tumor cells [13]. SCs interacting with tumor cells and the accompanying process of axonal sprouting provide the first access of cancer to nerves, which leads to neural dissemination at an early disease stage [14]. In pancreatic and colon cancer, SCs colonize neoplastic sites before the onset of tumor invasion, indicating their role in providing a pathway for tumor cell migration and invasion [13]. Furthermore, SCs can attract cancer cells to the perineural niche and enable adhesion of cancer cells to the nerves [15,16]. A recent study by Pascual et al. provided experimental evidence that a proregenerative state of tumor-activated SCs is related to metastasis initiation in mouse models of oral carcinogenesis and melanoma [17]. Another study demonstrated that SCs not only regulate the phenotype of tumor cells but also cancer-associated fibroblasts in the microenvironment of pancreatic ductal adenocarcinomas [18]. Taken together, SCs have crucial responsibilities in nerve–tumor interactions, indicating their potential role as cellular biomarkers and serving as drug targets for therapy.

Head and neck squamous cell carcinoma (HNSCC) is one of the most common cancers worldwide with high morbidity and mortality rates [19,20]. HNSCC is a remarkably heterogeneous disease, with tobacco, alcohol, and human papillomavirus (HPV) infection, especially HPV16, as major etiological risk factors [21]. Due to the abundant distribution of peripheral nerves in the head and neck region, HNSCC exhibits perineural invasion and intra-tumoral innervation to a greater extent than cancers originating from other anatomical sites [4,22]. Several recent landmark studies have provided compelling experimental and clinical evidence for a pivotal role of CNI in the pathogenesis of HNSCC and its potential as a promising drug target for anticancer therapy [6]. Amit et al. demonstrated in mouse models of oral cancer that the loss of the tumor suppressor p53 in tumor cells drives neuronal reprogramming through reduced levels of miR-34a. Moreover, sensory denervation or pharmacological blockade of adrenergic receptors inhibits tumor growth [23]. These findings from preclinical models were complemented by a retrospective analysis of tumor samples from oral cancer patients. This analysis not only confirmed an association between the local nerve density and the *TP53* mutation status, but also with poor clinical outcome. In another study, Restaino et al. identified synapse-like structures in HNSCC and demonstrated a higher electrical activity level in malignant as compared to benign or normal tissue. This finding suggests that intra-tumoral nerves establish functional connections to promote cancer progression [24].

Taken together, cancer neuroscience is emerging as an attractive new research avenue to better understand the underlying principles of HNSCC pathogenesis. Targeting the neural addiction of these tumors may provide an important new pillar for a more effective treatment of HNSCC patients [3,4,25]. To address these issues, there is an urgent need for new tools that allow quantitative assessment of peripheral nerve density in the tumor tissue from larger patient cohorts for which multi-omics and clinical data are available. The main objectives of this study were as follows: (i) the establishment of an SC-related gene set as a molecular surrogate for tumor nerve density, (ii) an integrative analysis of multi-omics data for tumors from TCGA-HNSC to elucidate differences in the mutational landscape and signaling networks as potential modulators of CNI, and (iii) the identification of potential vulnerabilities of tumors with functional neural circuits as drug targets for a more effective and less toxic treatment in HNSCC, but also in other cancer types.

## 2. Materials and Methods

### 2.1. Key Resources

Details of all publicly available data, patient cohorts, online tools, software, and algorithms used in this paper are listed in Appendix A. All data from TCGA are anonymized and the National Cancer Institute originally published the ethics and policies. The HNSCC patient cohort used in this study is from TCGA (TCGA-HNSC), consisting of 499 primary tumors as a training cohort and is complemented by validation cohorts for HNSCC (GSE117973, CPTAC-HNSC, GSE65858, and GSE39368) and for other tumor entities from TCGA (CESC, ESCA, LUSC, LUAD, PAAD, PRAD, and BRCA). The scRNA-seq data were downloaded from GSE182227, and the in silico drug screening analysis is based on CCLE data from the GDSC1-2 project and Oncopredict scores.

### 2.2. Establishment of the Schwann Cell-Related Gene Set

The Harmonizome tool represents a collection of processed datasets collected to serve and mine knowledge about genes and proteins from over 70 major online resources [26]. It provided 547 proteins co-occurring with the keyword “Schwann cells” in abstracts of biomedical publications from the “TISSUES Text-mining” Tissue Protein Expression Evidence Scores. The scores indicate the relative strength of the functional associations between genes/proteins and their attributes. Standardized scores are related to empirical *p*-values as abs (standardized values), and they are only available for initially continuous data sets. Based on the cutoff *p* < 0.05, *n* = 43 genes were selected to build an SC-related gene set.

### 2.3. Gene Set Variation Analysis (GSVA)

To integrate the selected SC-related 43-gene set into the computational analysis, we applied the Gene Set Variation Analysis (GSVA) package in RStudio (version 1.2.1335) using the Gaussian Kernel [27]. By this approach, we computed the distribution of enrichment scores for different gene sets, which are related to the peripheral nervous system (PNS), specifically SCs, based on RNA-seq data from the TCGA-HNSC cohort (*n* = 499). The GSVA provides scores to rank the patients according to the given gene sets. This model provided harmonized SC scores, which we used to stratify the tumors into three groups: SC^low^, SC^moderate^, and SC^high^. Eleven gene sets related to the peripheral nervous system (PNS) were selected from MSigDB [28] and applied to the GSVA model with the same purpose of ranking tumors from TCGA-HNSC. Three gene sets were selected from a study [29] that provided an unbiased single-cell transcriptomic characterization of the unperturbed rodent PNS, including independently confirmed markers of previously underestimated non-myelinating Schwann cells (nmSCs) and nerve-associated fibroblasts to compare the SC score with other published gene sets. We also calculated GSVA scores for an SC gene set provided by Panglao DB [30] based on single-cell RNA-seq data from mice and humans. For further validation on specific types of neurons, we included nine neural gene sets collected from Harmonizome [26], MSigDB [28], and Panglao DB [30]. The GSVA scores of the established gene sets were compared to the SC score using Spearman’s correlation.

### 2.4. Immunohistochemical Staining and Quantification

Paraffin-embedded tissue samples of primary HNSCCs from surgical resections of the GSE117973 cohort were provided by the Tissue Bank of the National Center for Tumor Diseases (NCT) Heidelberg, Germany, in accordance with the regulations of the tissue bank, the approval of the ethics committee of Heidelberg University (protocols: S-206/2011 and S-232/2022), and the written informed consent of all participants. This study was conducted in accordance with the Declaration of Helsinki. IHC staining was performed in accordance with Burkart et al. [31] using an anti-GAP43 antibody (ab75810, Abcam, Cambridge, UK). IHC-stained slides were scanned with a VENTANA DP 200 slide scanner (Roche, Mannheim, Germany), and to quantify the amount of GAP43-positive neuronal structures, the number of positive areas out of 10 was manually calculated.

### 2.5. Copy Number Alteration (CNA)

To explore the density and frequency of copy number gains and losses in groups with low, moderate, and high SC scores, we analyzed somatic CNA. Segmented data for TCGA-HNSC were downloaded from the “Firehose Broad GDAC”. The value of segment means was defined with greater than 0.2 as gain and less than −0.2 as loss, and depending on different *p*-value cutoffs, we obtained hotspot regions with the most significant CNV. Group comparisons were performed with differences in the frequency of specific events at each chromosomal location tested for significance using a two-tailed Fisher’s exact probability test with an accepted significance of the *p*-value at a defined percentage difference level using CoNVaQ [32]. Summary plots were visualized using IGV_2.8.0 software (Integrative Genomic Viewer_2.8.0).

### 2.6. Somatic Mutations

Mutation counts and candidate genes identified by MutSig 2.0 [33] for TCGA-HNSC were retrieved from cBioPortal (https://www.cbioportal.org/). Statistically significant differences between subgroups were determined by the chi-square test.

### 2.7. GSVA for Regulatory Networks and Oncogenic Pathways

Hallmark (H) category gene sets from the Molecular Signatures Database (MSigDB) were used, and enrichment scores were calculated using the GSVA package based on bulk RNA-seq (count data) from TCGA-HNSC. Statistical differences in GSVA scores between groups were analyzed using the Wilcoxon rank test.

### 2.8. Oncopredict

Cancer Cell Line Encyclopedia (CCLE) data were downloaded from the Dependency Map (DepMap) portal of the Broad Institute. Additionally, drug response data from the Sanger GDSC1 and GDSC2 drug screens were obtained from the same portal. The Oncopredict scores [34] were calculated using the R package Oncopredict (version 0.1) for TCGA-HNSC based on the CCLE RNA-seq and the IC50 drug response data of GDSC1-2. Statistical differences for the SC score (high vs. low) were calculated with the Wilcoxon rank test.

### 2.9. Single-Cell RNA-Seq Analysis

The single-cell RNA sequencing (scRNA-seq) data were downloaded from the Gene Expression Omnibus (GEO) [35]. The Unique Molecular Identifier (UMI) matrix was transformed to CPM (counts per million) by normalizing each gene by the total number of UMIs per sample. The CPM matrix was log2 transformed, as log2(CPM/10+1). The data were mean-centered by subtracting the average expression of each gene from all values of that gene.

### 2.10. Statistical Analysis

Statistical analysis was performed in R software (version 4.3.2), with *p*-values < 0.05 considered significant unless otherwise noted. Clinical data were analyzed by cross-tabulation, and significant enrichment was detected by the chi-square test using the R package (crosstable).

## 3. Results

### 3.1. Establishment of the Schwann Cell-Related 43-Gene Set

Assuming that the abundance of SCs serves as an accurate surrogate for the presence of peripheral nerves in a tumor, we used the Harmonizome online tool [26] to construct an SC-related 43-gene set. The gene set was extracted from 547 protein-coding genes that co-occurred with the keyword “Schwann cell” in abstracts of biomedical publications from the TISSUES Text-mining Tissue Protein Expression Evidence Scores dataset (cut-off *p*-value < 0.05, Appendix A). GSVA scores were calculated based on bulk RNA-seq data from TCGA-HNSC for the following: (i) the SC-related 43-gene set, (ii) eleven gene sets from MSigDB related to SCs or the peripheral nervous system, (iii) three gene signatures available for the unbiased single-cell transcriptomic characterization of the non-diseased rodents, and (iv) one gene signature for SCs from Panglao DB curated from mouse and human single-cell data. This analysis revealed a positive and significant correlation between the newly established SC-related 43-gene set and most of the other gene sets tested, with the exception of one gene set: GO_PERIPHERAL_NERVOUS_SYSTEM_MAINTENANCE (Figure 1A, Appendix A). The positive correlation was also confirmed with global gene expression data from four independent HNSCC cohorts (CPTAC-HNSC, GSE117973, GSE65858, GSE39368) and other solid tumors from TCGA with similar risk factors and histopathological characteristics (LUSC, CESC, and ESCA) or with well-established cancer–neuron interactions in the TME (LUAD, BRCA, PAAD, PRAD) [7,36] (Appendix A). To further substantiate the predictive accuracy of the SC-related 43-gene set, we inferred the enrichment of immune and stromal cells by deconvolution of gene expression data using xCell [37]. The comparison of these data with the GSVA scores for the SC-related 43-gene set confirmed a positive and significant correlation with the enrichment of astrocyte-like cells and neurons for TCGA-HNSC, independent HNSCC cohorts, and other solid tumors from TCGA (LUSC, CESC, ESCA, LUAD, BRCA, PAAD, PRAD) (Figure 1B, Appendix A). Notably, a significant and inverse correlation was also observed for several immune cells, particularly CD4+ memory T cells, CD8+ T cell memory, and T helper cells. For further validation, we extended this analysis by selecting eight gene sets related to specific sympathetic, parasympathetic, and neurogenesis-related gene sets from MSigDB, Panglao DB, and the Mayan Lab. A significant positive correlation was evident for all these gene sets, confirming the potential of the SC-related 43-gene set as a surrogate marker. However, we did not detect any obvious difference between sympathetic or parasympathetic gene signatures (Appendix A, Appendix A).

### 3.2. Association of the SC-Related 43-Gene Set with the Abundance of Peripheral Nerves in the TME

As a further proof of concept for the accuracy of the SC-related 43-gene set to predict peripheral neuron abundance in the TME, we examined FFPE tumor sections from a HNSCC cohort (*n* = 50) by IHC staining with an anti-GAP43 antibody. For this cohort, transcriptome data were available (GSE117973). GAP43 was selected because it is expressed in axons of peripheral neurons and activated SCs. The anti-GAP43 antibody had a good specificity with almost no positive staining in tumor cells or other stromal cells of the TME as compared to other antibodies for established marker proteins (Appendix A). Quantitative assessment of peripheral neurons adjacent to and within the tumor tissue showed an increase in the number of positive areas for HNSCC with a high GSVA score (SC^high^, *n* = 25) as compared to SC^low^ tumors (*n* = 25, Figure 1C,D). To further assess the quality of the predicted SC-related 43-gene set at the single-cell level, we calculated GSVA scores based on scRNA-seq data from a HNSCC cohort [38]. This analysis confirmed a significantly higher enrichment of the SC score in the mesenchymal subpopulation of fibroblast-like cells compared to all other cell types, including epithelial cells, endothelial cells, immune cells, lymphovascular cells, and myofibroblasts (Appendix A). These data indicated the absence or low expression of the SC-related 43-gene set in malignant epithelial cells.

In summary, the newly established SC score based on the SC-related 43-gene set provides an adequate tool to infer the abundance of peripheral nerves in the TME and to explore their mutual interaction with tumor cells during cancer development.

### 3.3. Association of the SC Score with Clinical Features and the Mutational Landscape

To explore potential differences in clinical and histopathologic characteristics among HNSCCs with different SC scores, we classified tumors from the TCGA-HNSC cohort into three groups: SC^low^ (lower quartile), SC^moderate^_,_ and SC^high^ (upper quartile). A cross-tabulation analysis revealed a highly significant enrichment of HPV16-positive OPSCC for the SC^low^ group, which was almost absent in the SC^high^ group (*p* = 2.33 × 10^−6^). Moreover, SC^high^ tumors were enriched for perineural invasion based on the histopathologic annotation (*p* = 2.50 × 10^−4^) and a recently established machine learning model (*p* = 1.00 × 10^−3^) (Appendix A) [39]. To further validate these findings, the SC scores based on single-cell RNA-seq data were compared with tumor cells from HPV-negative and HPV-positive tumors. This analysis confirmed a significantly higher SC score for single cells from HNSCC patients with HPV-negative tumors (Appendix A).

Regarding the mutational landscape, SC^high^ tumors exhibited a significantly lower proportion of global genomic alterations as compared to HNSCCs with low or moderate SC scores. Highly significant differences (*p* < 0.0005) were detected for copy number gains at chromosomes 3q and 9q and copy number losses at chromosomes 11q and 16q (Figure 2A,B). No statistically significant difference was evident in the total count of somatic mutations between HNSCCs with low, moderate, or high SC scores (Figure 2C). However, we identified several MutSig genes with significant differences in the relative frequency of somatic mutations (Figure 2D, Appendix A). Consistent with a recent study [23], we observed a significant (*p* = 0.0078) and gradual increase in the relative frequency of somatic *TP53* mutations with increasing SC scores. A similar pattern was also found for somatic *NOTCH1* mutations (*p* = 0.0262, Figure 2D). In contrast, a significant and gradual decrease in the relative frequency of somatic mutations with increasing SC scores was observed for *AGTR1* (*p* = 0.03), *RSRC1* (*p* = 0.02), and *PTEN* (*p* = 0.002). The higher frequency of somatic *PTEN* mutations in combination with *PIK3CA* copy number gain at chromosome 3q26 for SC^low^ as compared to SC^high^ HNSCC suggested an inverse association between PI3K pathway activity in cancer cells and peripheral nerve abundance in the TME.

### 3.4. Association of the SC Score with Gene Regulatory Networks and Oncogenic Pathway Activity

To further support the notion that PI3K pathway activity hinders tumor-related neurogenesis and to identify other underlying molecular principles of the reciprocal cancer–neuron interaction, we calculated GSVA scores for hallmark gene sets of the MSigDB (category H) based on bulk RNA-seq data from TCGA-HNSC (Figure 3A, Appendix A). Consistent with differences in the mutational landscape, significantly higher GSVA scores were detected for the P53 pathway (*p* = 4.9 × 10^−3^), PI3K-AKT-MTOR (*p* = 2.01 × 10^−3^), and MTORC1 signaling (*p* = 3.91 × 10^−9^) in SC^low^ tumors as compared to SC^high^ counterparts (Figure 3A,B). Other gene sets with a higher GSVA score for SC^low^ HNSCCs resembled processes of metabolism and cell cycle progression, such as MYC or E2F target genes, oxidative phosphorylation, DNA repair, and G2M checkpoint (Figure 3A,C and Appendix A). In contrast, top-ranked gene sets with higher GSVA scores for SC^high^ HNSCCs were associated with well-established oncogenic processes and pathways, such as epithelial–mesenchymal transition (EMT) and Hedgehog and TGF-β signaling (Figure 3A,D). These associations were independent of the HPV16 status in TCGA-HNSC (Appendix A) and were also confirmed for the CPTAC-HNSC cohort (Appendix A). Significantly higher GSVA scores for PI3K-AKT-MTOR and MTORC1 signaling in SC^low^ tumors and higher GSVA scores for gene sets related to TGF-β signaling and EMT in SC^high^ tumors were also observed in other solid tumors from TCGA, such as CESC and BRCA (Appendix A). Finally, we confirmed significantly higher TGF-β pathway activity for SC^high^ tumors for TCGA-HNSC with PROGENy (Appendix A, Appendix A). This activity may contribute to a suppressed tumor immune microenvironment (TIME) with a reduced number of TILs, as detected previously (Appendix A). In summary, the analysis of GSVA scores for hallmark gene sets confirmed higher PI3K-AKT-MTOR signaling pathway activity in the low SC group, independent of HPV16 status. These data further support a negative impact of PI3K pathway activity on peripheral nerve abundance within the TME.

### 3.5. Context-Dependent Impact of PI3K Pathway Activity

To further explore the impact of PI3K pathway activity on peripheral nerve abundance in the context of somatic *TP53* mutations, we stratified tumors from TCGA-HNSC into four groups: (i) tumors without somatic *TP53*, *PTEN*, or *PIK3CA* mutations and without *PIK3CA* amplification (*TP53^wt^/PTEN-PIK3CA^wt^*); (ii) tumors with somatic *TP53* mutations but without somatic *PTEN* or *PIK3CA* mutations or *PIK3CA* amplification (*TP53^mut^/PTEN-PIK3CA^wt^*); (iii) tumors without somatic *TP53* mutation but with somatic *PTEN* or *PIK3CA* mutations or *PIK3CA* amplification (*TP53^wt^/PTEN-PIK3CA^mut^*); or (iv) tumors with somatic *TP53* mutations and with somatic *PTEN* or *PIK3CA* mutations or *PIK3CA* amplification (*TP53^mut^/PTEN-PIK3CA^mut^*). Consistent with our hypothesis that PI3K pathway activity is impaired with peripheral nerve abundance, SC scores were significantly lower in *TP53^wt^/PTEN-PIK3CA^mut^* HNSCC as compared to all other groups (*p* = 0.01, Figure 4A). However, this difference was not evident in the presence of somatic *TP53* mutations, suggesting that the impact of PI3K pathway activity on the abundance of peripheral neurons is context-dependent and dominated by the *TP53* status. A similar mode of regulation was also evident for neurotrophic factors, such as NGF (Figure 4B).

To further support this concept, we examined the expression of miR-34a, which was recently reported to be a p53-regulated target during neuronal reprogramming in HNSCC [23]. As expected, miR-34a showed a significantly lower expression in SC^high^ as compared to other tumors for TCGA-HNSC (*p* < 0.01, Figure 4C). A significant difference in miR-34a expression related to the SC score was also confirmed in other solid tumors from TCGA, such as LUSC, ESCA, CESC, and PAAD (Appendix A). Regarding the context-dependent role of PI3K pathway activity, the highest miR-34a transcript levels were detected in *TP53^wt^/PTEN-PIK3CA^m^*^ut^ HNSCC. In contrast, its expression was significantly lower in the presence of somatic *TP53* mutations (Figure 4D). A reverse mode of regulation was also detected for TGF-β signaling and a significant difference was observed between *TP53^wt^/PTEN-PIK3CA^wt^* and *TP53^mut^/PTEN-PIK3CA ^mut^* tumors from TCGA-HNSC (Figure 4E).

Next, we calculated the SC score based on RNA-seq data from 4NQO-induced tongue tumors of genetically engineered mice carrying a gain-of-function (GOF) allele for *Pik3ca* alone or in combination with a *Trp53* mutation [40]. The analysis revealed a significantly higher SC score for tongue tumors from *Trp53^mut^* mice with or without *Pik3ca^GOF^* as compared to wild-type (WT) controls or *Pik3ca^GOF^* counterparts (Figure 4F). Tongue tumors from this animal study also confirmed a significantly higher Ngf expression in the presence of *Trp53* mutation compared to WT controls and *Trp53^wt^/Pik3ca^GOF^* mice (Figure 4F).

Finally, we analyzed the PI3K pathway activity and *TP53* mutation status for tumors which were stained with the anti-GAP43 antibody. As expected, GSVA scores for the SC-related 43-gene set were higher in *TP53* mutant tumors compared to non-mutant counterparts (Figure 4G). PI3K pathway activity was inferred from the transcriptome data using PROGENy. The inferred activity was lowest for tumors without *TP53* mutation and with a high abundance of peripheral nerves as assessed by GAP43 immunostaining (Figure 4G). Collectively, these data confirmed that the impact of increased PI3K pathway activity on the abundance of peripheral neurons within the TME is context-dependent and dominated by the *TP53* status.

### 3.6. Identification of Vulnerabilities by in Silico Drug Screening

To identify potential vulnerabilities for tumor subgroups with either SC^high^ or SC^low^ phenotypes, we used Oncopredict [34] to impute sensitivity to compounds tested in the GDSC project (Appendix A). This approach revealed potential sensitivities of SC^high^ tumors to EGFR inhibitors (e.g., CANERTINIB, AST-1306) and MEK inhibitors (e.g., REFAMETINIB), whereas SC^low^ tumors were particularly sensitive to drugs targeting the cell cycle or DNA topoisomerases (e.g., CAMPTOTHECIN, IRINOTECAN, TENIPOSIDE, and TOPOTECAN) (Appendix A). The latter finding was consistent with the enrichment of gene sets related to cell cycle progression and DNA repair in the SC^low^ tumors (Figure 3A,B, Appendix A) and indicates a therapeutic potential of topoisomerase inhibitors for HNSCC patients with this type of tumor.

## 4. Discussion

High density of peripheral nerves and their functionality in the TME have been associated with poor prognosis in several cancers, including HNSCC [8,22,41]. However, our understanding of the underlying molecular and cellular principles that promote mutual communication between cancer cells and peripheral nerves remains limited [4]. This is, at least in part, due to the technical challenge of reliably assessing peripheral nerve abundance in tumor samples from larger patient cohorts for which multi-omics data are available. In addition, most research efforts focus on the impact of different neuronal types, e.g., sympathetic, parasympathetic, or sensory neurons, in carcinogenesis, generally ignoring neuroglia such as SCs [42]. SCs are able to modulate various immune cells and inflammatory processes and possess regenerative capacities. Their broad abundance throughout the body makes them a perfect target for malignant cells during the initial stages, progression of cancer development, and the formation of the tumor microenvironment [42]. The reprogramming of SCs occurs in cancer and is associated with a variety of effects, including cancer growth, cancer migration and invasion, immune regulation, and pain. Immune regulation and pain transmission may be controlled directly by the cancer cells or indirectly via the reprogrammed SCs in the microenvironment [43]. Several studies showed that SCs in the TME attract various subsets of immune regulators and enhance their ability to suppress effector T cells [42].

In the present study, we established an SC-related 43-gene set as an accurate surrogate for the presence of peripheral nerves in the TME of HNSCC and classified TCGA-HNSC tumors into three groups based on low, moderate, and high SC scores. The predictive power of the SC score was confirmed by gene expression data from independent HNSCC cohorts and other solid tumors from TCGA with similar risk factors and histopathologic characteristics (LUSC, CESC, and ESCA) or with well-established cancer–neuron interactions in the TME (LUAD, BRCA, PAAD, PRAD) [7,36]. A landmark study by Amit et al. identified loss of p53 function as an important regulator of neuronal reprogramming in HNSCC, thereby promoting tumor development and malignant progression [23]. *TP53* is the most frequently mutated tumor suppressor gene in HNSCC and affects multiple aspects during tumorigenesis, including modulation of the TME [21]. Mechanistically, loss of p53 was associated with loss of miR-34a expression, which determines the fate of cancer-associated neurons [23]. MiR-34a is a well-known tumor suppressor miRNA that is lost or downregulated in a variety of tumors. Moreover, the reintroduction of miR-34a mimics was found to inhibit cancer cell growth both in vitro and in vivo. In nasopharyngeal carcinoma, miR-34a was identified as a critical inhibitor of EMT and metastasis [44,45]. Consistent with these findings, we confirmed a higher frequency of somatic *TP53* mutations and reduced miR-34a expression in SC^high^ HNSCC, while p53 pathway activity and miR-34a expression were higher in SC^low^ tumors. Interestingly, a significant difference in miR-34a expression related to the SC score was also observed in other solid tumors from TCGA, such as LUSC, ESCA, CESC, and PAAD. This finding raises the intriguing question of whether our newly established SC score can be used to identify tumors with *TP53* mutations and a high local nerve density in the TME. These cases may benefit from miR-34a mimetics in combination with currently established treatment regimens.

Another important finding of this study is the potential role of the PI3K-MTOR pathway in the regulation of nerve density within the TME of HNSCC. The analysis of gene regulatory networks and oncogenic pathway activities revealed significantly higher GSVA scores for PI3K-AKT-MTOR and MTORC1 signaling in SC^low^ HNSCC. These associations were at least in part due to the higher frequency of somatic *PTEN* mutations and *PIK3CA* copy number gain at chromosome 3q26 in these tumors as compared to the SC^high^ counterparts. Genomic alterations and somatic mutations in key hubs of the PI3K signaling pathway are common in HPV16-associated OPSCC and are associated with the activation of MTOR. This specific difference in the mutational landscape, in combination with the lack of somatic *TP53* mutations, may explain the reduced abundance of peripheral nerves in the TME of HPV16-positive OPSCC. Accordingly, HPV16-positive OPSCC do not only exhibit lower SC scores, but also have reduced expression of synaptic markers and electrical activity within tumors and a lower frequency of PNI [24,39]. However, it is worth noting that the potential impact of PI3K-MTOR signaling on neuro- and/or axonogenesis appears to be context-dependent and influenced by the *TP53* mutation status. This assumption is supported by the low SC scores in *TP53^wt^/PTEN-PIK3CA^mut^* tumors from HNSCC patients and chemically-induced tongue tumors from *Trp53^wt^/Pik3ca^GOF^* mice. Low SC scores were not observed for tumors with somatic *TP53* mutations despite the presence of *PTEN* or *PIK3CA* alterations. Recent studies have shown that cancer cells express neurotrophic markers such as NGF, BDNF, and GDNF and release axon guidance molecules such as Ephrin B1 to promote axonogenesis, neurogenesis, and neuronal reprogramming [6]. Indeed, NGF expression was lowest in tumors from HNSCC patients and tongue tumors from animals with mutations in the PI3K pathway but without *TP53* mutations. These findings suggest that the effect of the PI3K-MTOR pathway activity on the abundance of peripheral neurons in the TME is triggered, at least in part, by the altered release of NGF, which is dependent on the TP53 status. However, the underlying molecular principles remain elusive and need to be addressed in future studies.

In the context of the tumor immune phenotype, TGF-β signaling is known to suppress the function of adaptive and innate immune cells [46,47]. Previous studies have reported a close interaction between peripheral neurons, SCs, and immune cells, which are mediated by soluble factors, including TGF-β [48]. For example, a recent study provided compelling experimental evidence that neural involvement enhanced tumor aggressiveness by upregulating TGF-β signaling and PD-L1 expression in OSCC. Moreover, denervation of OSCC inhibited tumor growth, which was accompanied by reduced TGF-β signaling, enhanced CD8^+^ T cell activity, and improved efficacy toward anti-PD-1 immunotherapy [49]. Consistent with this study, our data demonstrated a positive correlation between a high SC score and TGF-β signaling. This finding indicates an immunosuppressive phenotype for SC^high^ tumors based on the reduced levels of specific tumor-infiltrating lymphocytes as determined by xCell. In contrast, SC^low^ tumors demonstrated an immunoactive TME by enrichment of CD8^+^ T cells, CD4^+^ T cells, and T helper cells. It is also noteworthy that HPV16-positive OPSCCs were almost absent in the SC^high^ group. OPSCCs arise from anatomical sites with lymphoid tissue, especially tonsils, which could also explain a higher density of different immune cells in the TME. However, the presence and activity of specific immune cell subsets may also have a negative impact on peripheral nerve density by inhibiting neurogenesis and axonogenesis in an immunoactive environment. These findings highlight promising therapeutic targets in the context of cancer–neuron–immune crosstalk and further investigation in co-culture models using cancer cells, SCs, and immune cells will be important as a proof of concept for future translation into clinical application.

Finally, we explored the potential vulnerabilities for HNSCC with SC^high^ or SC^low^ phenotypes as a solid basis for further testing in appropriate preclinical models and long-term translation into clinical trials. Our in silico drug screening predicted a higher sensitivity of SC^low^ tumors to drugs targeting DNA topoisomerases, such as Camptothecin, Irinotecan, Teniposide, and Topotecan. Inhibitors of the DNA topoisomerase 1 (TOP1) and 2 (TOP2A and TOP2B) have shown considerable potential as therapeutic agents against cancers, including HNSCC. However, their unfavorable pharmacokinetic profiles and off-target toxicities have limited their clinical use [50,51]. In HNSCC, a phase I/II prospective trial of triple combination therapy with Cisplatin, Tegafur/Uracil, and Irinotecan showed that this treatment was well tolerated and effective in selected patients [52]. These data suggest that a better molecular stratification may improve prognosis and quality of life for individual patients with recurrent/metastatic HNSCC. To maximize the therapeutic potential of DNA topoisomerase inhibitors, combination therapies co-targeting oncogenic pathways have been proposed to induce synthetic lethality for more thorough anti-tumor activity [53]. As our GSVA identified enrichment of MYC and E2F target genes for SC^low^ tumors, co-targeting of these oncogenic pathways in combination with DNA topoisomerase inhibitors may be a viable option for these HNSCCs. In addition, the list of top-ranked candidates with a higher sensitivity for SC^high^ HNSCC included EGFR inhibitors (e.g., Canertinib, AST-1306) and MEK inhibitors (e.g., Refametinib). Again, a more detailed analysis in appropriate preclinical models is needed to provide the proof of concept that inhibition of the EGFR-MEK pathway has an impact on CNI and resembles an effective treatment for SC^high^ HNSCC.

We are aware that our study is not without limitations. The limitations are the retrospective study design and that most conclusions are derived from bioinformatics analysis of bulk RNA-seq data from TCGA and other publicly available resources. Future studies have to focus on prospectively collected tumor samples, for which both the assessment of the newly established SC-related 43-gene set and multiplex immunostaining on tumor sections will be feasible. Furthermore, a more precise definition of candidate genes for several subsets of SCs and peripheral nerves will be necessary to better characterize their abundance, activity, and regulation in the TME at the single-cell level (e.g., by the analysis of single-cell RNA-seq and spatial transcriptomics data). Finally, the impact of predicted oncogenic pathways in tumor cells and their context-dependent function in the complex interplay between cancers and key components of peripheral nerves in the TME must be experimentally confirmed by sophisticated preclinical 3D coculture and adequate in vivo models. Expected data will provide the proof of concept for the most promising drug targets and their translation into clinical trials.

## 5. Conclusions

In this study, the presence of peripheral nerves in the TME across HNSCC and other tumor entities was predicted by a SC-related 43-gene set. The newly established SC score provides an adequate tool to infer the abundance of peripheral nerves in the TME and to explore their mutual interaction with tumor cells during cancer development. Based on the comparison of tumors with either a low or high SC score, we gained new clinical and biological insights which allowed us to predict the potential vulnerabilities of tumor subgroups with variable abundance of peripheral nerves in the TME. This approach revealed potential sensitivities of SC^high^ tumors against inhibitors targeting EGFR-MEK signaling, whereas SC^low^ tumors appear to be particularly sensitive to drugs targeting DNA topoisomerases. The critical role of the PI3K-MTOR pathway in cancer cells, which is associated with the altered expression of important regulators of neurogenesis, axonogenesis, and neuronal reprogramming, is another key finding of our study. However, the impact of PI3K-MTOR pathway activity on the abundance of peripheral neurons within the TME is context-dependent and affected by the TP53 status. These findings may pave the way for new therapeutic strategies and impart proof of concept in appropriate preclinical models.

## Figures and Tables

**Figure 1 cells-13-01033-f001:**
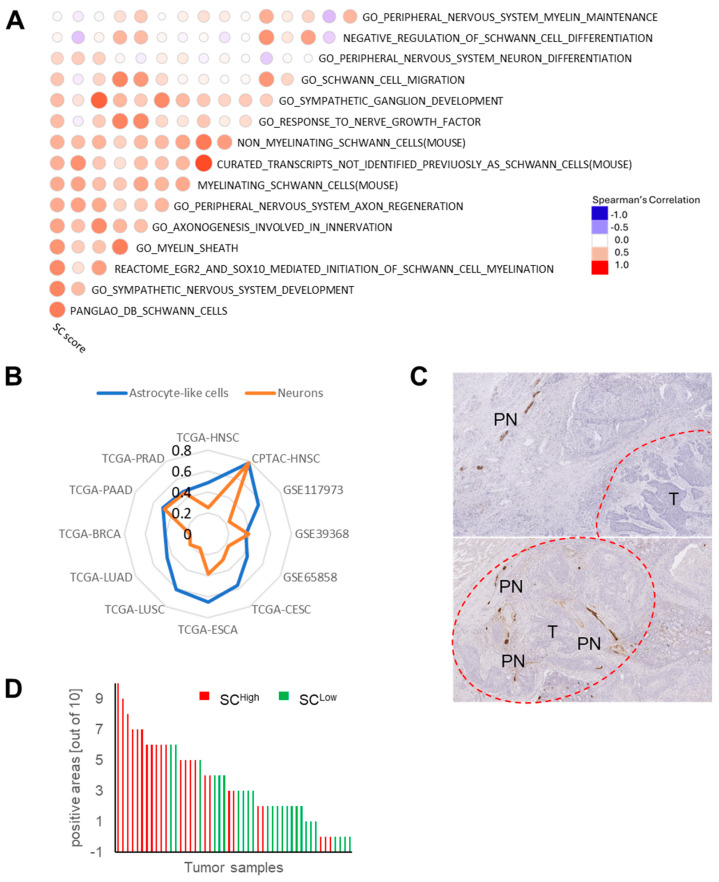
Establishment of the SC-related 43-gene set as a surrogate for peripheral nerve abundance in the TME of HNSCC. (**A**) Spearman’s correlation coefficient matrix for GSVA scores of the SC-related 43-gene set and indicated gene sets related to the peripheral nervous system based on bulk RNA-seq data from TCGA-HNSC. (**B**) Spider plot showing the correlation coefficients comparing the GSVA scores of the SC-related 43-gene set and scores for astrocytes (blue color) and neurons (orange color) based on xCell deconvolution for TCGA-HNSC, four independent HNSCC cohorts, and other solid tumors from TCGA. (**C**) Representative images of an IHC staining with an anti-GAP43 antibody show a positive staining (brown signal) in peripheral nerve fibers (PN) adjacent to and within the tumor area (T, marked by a dashed red line) of FFPE tumor sections. Histological staining with hematoxylin to visualize tissue architecture. (**D**) Bar plot shows quantitative assessment of positive areas of peripheral neurons adjacent to and within the tumor and demonstrates an increase in the number for HNSCC with a high SC score (SC^high^, red, *n* = 25) as compared to a low SC score (SC^low^, green, *n* = 25).

**Figure 2 cells-13-01033-f002:**
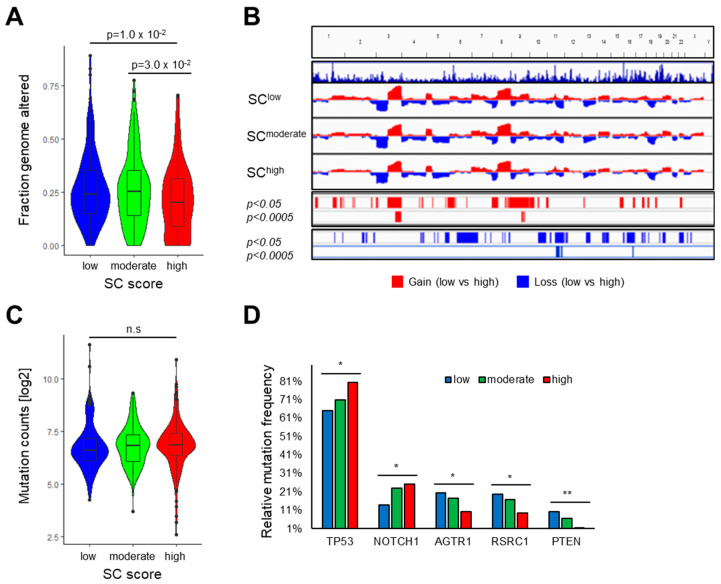
Differences in the mutational landscape for tumors from TCGA-HNSC with low, moderate, and high SC scores. (**A**) Violin plot shows statistically significant differences in the fraction of genome altered for HNSCC with low, moderate, and high SC scores. (**B**) Frequency plots summarize copy number gains (red) and losses (blue) and indicate hotspot regions with significant differences between tumors from TCGA-HNSC with high and low SC scores. (**C**) Violin plot shows no statistically significant difference in mutation counts for HNSCCs with low, moderate, and high SC scores. (**D**) Bar plot illustrates MutSig genes (*TP53*, *NOTCH1*, *AGTR1*, *RSRC2*, *PTEN*) with statistically significant differences in somatic mutation frequency among HNSCCs with low, moderate, and high SC scores. Statistical differences in the fraction of genome altered and mutation counts were determined by ANOVA-Tukey HSD, while the significant differences in relative mutational frequency were determined by chi-square test with * *p* < 0.05, ** *p* < 0.005 and n.s. = not significant.

**Figure 3 cells-13-01033-f003:**
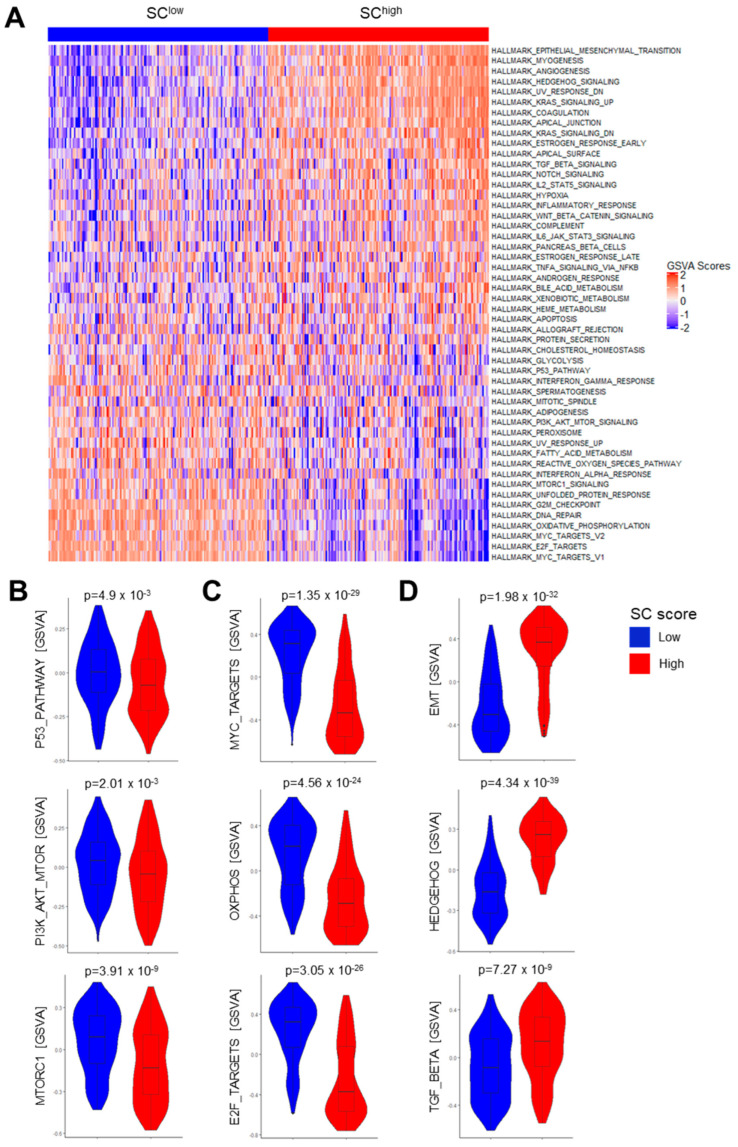
Differences in gene regulatory networks and oncogenic pathway activity associated with different SC scores for tumors from TCGA-HNSC. (**A**) Heatmap showing an unsupervised hierarchical cluster analysis based on GSVA scores of top-ranked MSigDB hallmark gene sets with significant differences between tumors with high or low SC score. (**B**–**D**) Violin plots show statistically significant differences between tumors with high or low SC scores for indicated gene sets in TCGA-HNSC. Statistical significance was determined using the Wilcoxon rank test.

**Figure 4 cells-13-01033-f004:**
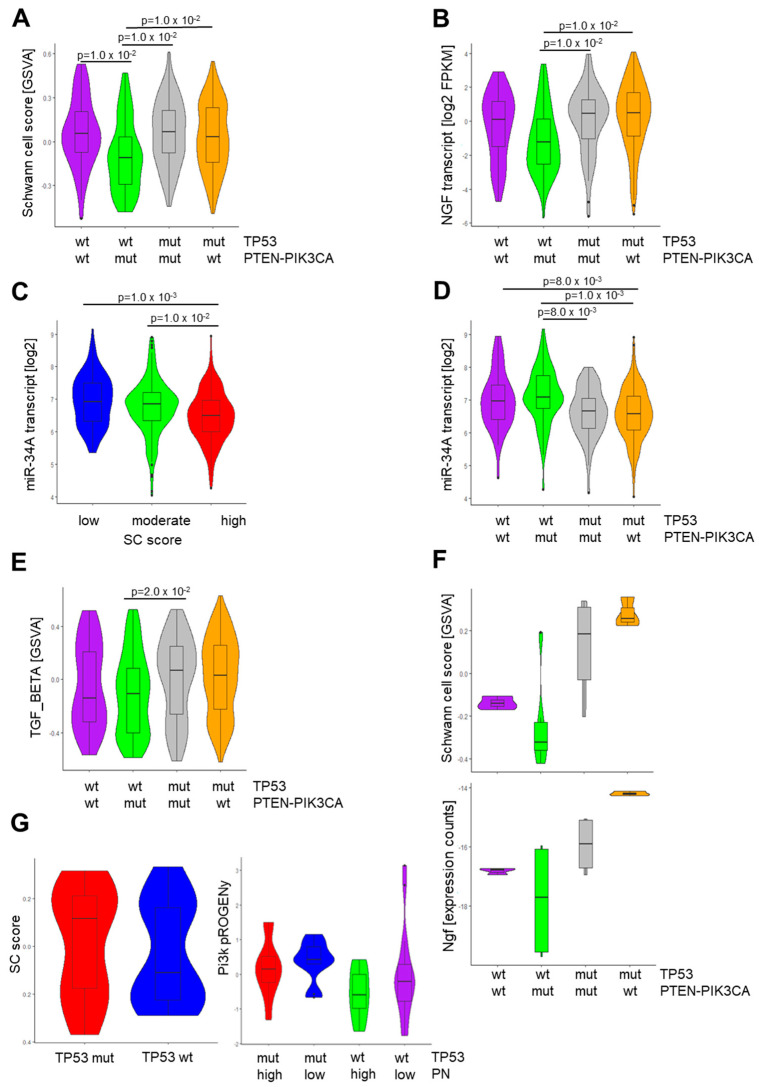
Association of the SC score with PI3K pathway activity, NGF, and miR-34a expression. Violin plots showing differences between indicated mutation subgroups of TCGA-HNSC for the SC score (**A**), NGF transcript levels (**B**), miR-34a transcript levels (**D**), and GSVA scores for TGF-β signaling (**E**). (**C**) Violin plot showing statistically significant differences in miR-34a transcript levels for TCGA-HNSC tumors with low, moderate, and high SC scores. (**F**) Violin plots illustrating SC scores (top) and Ngf transcript levels (bottom) for 4NQO-induced tongue tumors from mice with indicated genotypes. (**G**) Violin plots showing differences in the SC score between tumors from GSE117973 with or without *TP53* mutation (**left**) and for the PI3K PROGENy score for tumors’ *TP53* mutation status and the abundance of peripheral nerves as assessed by GAP43 immunostaining (**right**).

## Data Availability

The data generated in this study are available within the article and its Appendix A files. Publicly available data generated by others were used and listed in Appendix A.

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
