# Peer review of "Context-Dependent Regulation of Peripheral Nerve Abundance by the PI3K Pathway in the Tumor Microenvironment of Head and Neck Squamous Cell Carcinoma"

_cells, 2024, doi:10.3390/cells13121033_

Round 1

Reviewer 1 Report

Comments and Suggestions for Authors

This report is a bioinformatic analysis of a 43 Schwann cell (SC) gene signature, designed by the authors and based on citations in publications, in head and neck squamous cell carcinoma. It leads to the discovery of the association of the SC signature with pathways such TGFbeta signaling and PI3K-MTOR activity within subsets of head and neck squamous cell carcinoma.

Introduction:

Since the paper is on SC rather than neurons, the introduction should provide proper references for SC finding in cancer rather than review papers. Missing pieces of literatures on the role of Schwann cells in cancer include publications from Wong lab ((JCI, Cancer Discovery), Bunimovich lab and others (Xue et al Nat Comm 2023, Pascual et al Nature 2021, Demir lab)

Results:

Authors mentioned that SC signatures are good to assess tumor innervation. Can they show results on neuron gene sets? And provide information on types of innervation using Sympathetic, parasympathetic, sensory signatures.

Discussion:

The papers reads as if the SC signatures are used as a surrogate for neuron signatures. The importance of the role of SC in cancer is not discussed in the paper. The role in immune function and role in cancer invasion should be discussed.

Also, it would be appropriate to mention that other studies reported a correlation of SC signatures scores with pathways mentioned in this study including TGF-beta signaling, PI3K-mTOR signaling and immune related pathways. See PMID: 35881881

There must be some mistakes on reporting the source of some signatures:

Table S1 indicates that Schwann cells related gene sets (Rodents) corresponds to publication PMID 32051587, which must be an error since this paper does not report such gene sets.

Thus, we do not know what the source of the following gene sets and what genes composed them:

MYELINATING_SCHWANN_CELLS (MOUSE)

CURATED_TRANSCRIPTS_NOT_IDENTIFIED_PREVIUOSLY_AS_SCHWANN_CELLS (MOUSE)

NON_MYELINATING_SCHWANN_CELLS (MOUSE)

It would be a plus. if the authors can distinguish between myelin and non-myelin SC too.

Reviewer 2 Report

Comments and Suggestions for Authors

The manuscript is only clinical statistical analysis using clinical sample. Nobody knows the reason why the effect of PI3kinase activity on the abundance of peripheral neurons is negative. The manuscript needs more depth.

Comments on the Quality of English Language

English very difficult to understand

Reviewer 3 Report

Comments and Suggestions for Authors

This article was used SC (Schwann cells)-related 43-gene as the GSVA scores to subgroup the HNSCC as low, moderate and high of SC as the peripheral nerve index in the tumor microenvironment. But only the Figure 1C and 1D is the web lab data used the anti-GAP43 antibody show a positive staining as the peripheral nerve fibers with the positive correlation with GSVA scores in clinical samples. All other data were dry lab database analyze for genes mutated status and targeting drug prediction according to the GSVA scores. It should be used the clinical samples for staining the p53 and PI3K pathway for low and high SC clinical samples. There is p53 mutated specific antibody can help to verify the p53 status for IHC stain. And many p53 or PI3K pathway downstream genes specific antibodies also can be used in IHC staining for this research. 

Reviewer 4 Report

Comments and Suggestions for Authors

The manuscript entitled ‘Context-dependent regulation of peripheral nerve abundance 2 by the PI3K pathway in the tumor microenvironment of head 3 and neck squamous cell carcinoma’ is well organized, and with the comments below, the paper can undergo minor revision.

1.      The authors are suggested to replace the abbreviations with the full form in the keywords.

2.      It is recommended that a couple of original research pieces be added to the introduction.

3.      Certain sentences written are complicated and long. It is suggested that the sentence be broken and made short. E.g., ‘To integrate the selected SC-related 43-gene…’

4.      The conclusion can be enriched with more results.

5.      The importance of PI3K/Akt/mTOR pathway in cancer cell survival is already well understood. How is your finding different from this concept?

Comments on the Quality of English Language

It is written with very complex and long sentences, and it is suggested to make it simple and small.

Round 2

Reviewer 3 Report

Comments and Suggestions for Authors

It’s still highly recommended to co-stain anti-GAP43 and anti-p53 antibodies on your own IHC samples to completing this research. Only needing two antibodies (pab1620 and pab240) are enough to distinguish WT or mutation of p53 for IHC samples (ex: PMID: 8319499 & 10810396 ). 
